# Austenite Decomposition of a Lean Medium Mn Steel Suitable for Quenching and Partitioning Process: Comparison of CCT and DCCT Diagram and Their Microstructural Changes

**DOI:** 10.3390/ma15051753

**Published:** 2022-02-25

**Authors:** Michal Krbata, Daniel Krizan, Maros Eckert, Simone Kaar, Andrej Dubec, Robert Ciger

**Affiliations:** 1Faculty of Special Technology, Alexander Dubcek University of Trenčín, 911 06 Trenčín, Slovakia; michal.krbata@tnuni.sk (M.K.); robert.ciger@tnuni.sk (R.C.); 2Research and Development Department, Business Unit Coil. Voestalpine Steel Division GmbH, Voestalpine-Strasse 3, A-4020 Linz, Austria; daniel.krizan@voestalpine.com (D.K.); simone.kaar@voestalpine.com (S.K.); 3Faculty of Industrial Technologies, Alexander Dubcek University of Trenčín, 020 01 Puchov, Slovakia; andrej.dubec@tnuni.sk

**Keywords:** dilatometry, martensite, austenitization temperature, lean medium Mn steel, CCT, DCCT

## Abstract

The present work deals with the dilatometric study of a hot-rolled 0.2C3Mn1.5Si lean medium Mn steel, mainly suitable for the quenching and partitioning (Q&P) heat treatment in both hot-rolled or cold-rolled condition, subjected to a variation of austenitization temperature. These investigations were performed in a temperature range of 800–1200 °C. In this context, the martensite transformation start temperature (Ms) was determined as a function of austenitization temperature and in turn obtained prior austenite grain size (PAGS). The results show rise in prior austenite grain size due to increasing austenitization temperature, resulting in elevated Ms temperatures. Measured dilatation curves were confronted with the metallographic analysis by means of scanning electron microscopy (SEM). The present paper also focuses on the construction of a continuous cooling transformation (CCT) and deformation continuous cooling transformation (DCCT) diagram of the investigated lean medium Mn steel in a range of cooling rates from 100 to 0.01 °C/s and their subsequent comparison. By comparing these two diagrams, we observed an overall shift of the DCCT diagram to shorter times compared to the CCT diagram, which represents an earlier formation of phase transformations with respect to the individual cooling rates. Moreover, the determination of individual phase fractions in the CCT and DCCT mode revealed that the growth stage of ferrite and bainite is decelerated by deformation, especially for intermediate cooling rates. Microstructural changes corresponding to cooling were also observed using SEM to provide more detailed investigation of the structure and present phases identification as a function of cooling rate. Moreover, the volume fractions obtained from the saturation magnetization method (SMM) are compared with data from X-ray diffraction (XRD) measurements. The discussion of the data suggests that magnetization measurements lead to more reliable results and a more sensitive detection of the retained austenite than XRD measurements. In that regard, the volume fraction of retained austenite increased with a decrease of cooling rate as a result of larger volume fraction of ferrite and bainite. The hardness of the samples subjected to the deformation was slightly higher compared to non-deformed samples. The reason for this was an evident grain refinement after deformation.

## 1. Introduction

Dilatometric and plastometric analysis using modern dilatometers is currently a widespread method for the evaluation of microstructural changes and selected properties (hardness, magnetic properties) of metallic materials [1]. In the case of dilatometric analysis of steels, current research focuses mainly on the study of phase transformations such as austenitizing during heating, but also the transformation of austenite during a given cooling stage [2,3].

The lean medium Mn steel is a middle alloyed steel based on a C-Mn-Si alloying concept. In the hot-rolled and cold-rolled condition, when applied a special Q&P heat treatment, a specific combination of mechanical properties and strengths higher than 1100 MPa can be achieved [4,5].

Standard medium Mn alloyed steels with Mn contents between 3 and 12 wt.% belong to the 3rd generation of advanced high strength steels (AHSS) used in the automotive industry [6]. The microstructure of these steels consists of metastable retained austenite (about 30 vol.%) in the ferritic matrix. Metastable austenite transforms to martensite during forming processes of automotive components such as deep drawing. This mechanism, well-known as the transformation-induced plasticity (TRIP) effect, results in excellent combination of mechanical properties with ultimate tensile strength and total elongation (UTSxTE) exceeding even 30,000 MPa% [7,8]. These steels are designed to produce high-strength automotive components with very complex shapes produced by deep drawing such as pillars and reinforcements with a purpose to protect automobile passengers in an event of crash [9,10].

Lean medium Mn steel are a subgroup of conventional medium Mn steels with a reduced Mn content in a range of 3 to 4.5 wt.% and C content with an upper limit of approximately 0.2 wt.%. Standard medium Mn steel is treated by intercritical annealing to achieve 70 vol.% of ferrite and 30 vol.% of retained austenite in the final microstructure. However, heat treatment of lean medium Mn steel is based on a different annealing process named quenching and partitioning (Q&P), consisting of quenching and following partitioning of C from martensite to retained austenite [11].

These steels are still in the development phase and the world’s leading steel mills, are currently carrying out the first casting tests in production [12]. The Q&P process consists of the following basic steps: The steel is heated to a temperature in the stable austenite region (above Ac_3_ line). After some holding period at that temperature (named as annealing temperature—T_γ_), very fast quenching takes place at the temperature below M_s_, which results in the microstructure consisting of high fraction of retained austenite and martensite (T_Q_—Tquech). Next step is reheating up to the partitioning temperature (usually above Ms) and holding period at the temperature, where C is redistributed from martensite to austenite (T_P_—T partitioning). As a result of T_P_, the martensite is tempered and austenite becomes enriched in C. The final structure consists of about 10 ÷ 15 vol.% of metastable retained austenite which allows the transformation-induced plasticity steel (TRIP) effect based on deformation-induced transformation of austenite to martensite [13]. Then, the final mechanical properties of such a treated steel are tensile strength above 1100 MPa and higher elongation as compared with conventional quenched and tempered steels (Q&T). In this case, the product of UTS and TE of approximately 25,000 MPa% is achievable. These steels will be used in the near future mainly as a side and anti-intrusion car body reinforcement structures (A, B, and C pillars, fenders, etc.,) [14,15].

The studies of Franceschi M. et al. [16,17] dealt with the influence of different or multi-stage tempering on the resulting mechanical and corrosion-resistant properties of bainitic steel. It is clear from their results that an increase in the austenitization temperature leads to an increase in the volume fraction of retained austenite. Dilatometric analysis of austenite decomposition in undeformed and deformed low carbon structural steel was discussed by Morawiec M. et al. [18,19]. Their results show that the hot deformation performed before cooling increases the diffusion rate of elements and highly influences the phase transformation kinetics. The plastic deformation substantially enhanced grain refinement in the whole range of applied cooling rates. The hardness of the steel increases along with the increasing cooling rate, due to progressive grain refinement and lower transformation start temperatures, also inducing smaller grain sizes. Morales-Rivas, L. [20] dealt with the investigation of the technological aspects of AHSS. The microstructure evolution of high-strength steels during hot and cold rolling and cooling from an austenitizing temperature were investigated [21,22] in findings of the paper. Zhao et al. [23] reported that the refinement and strengthening of AHSS were achieved by thermomechanical processing. Sugimoto et al. [24] stated that the re-fined, recrystallized ferrite in an annealed martensite matrix was obtained when hot forging was conducted on the TRIP-aided annealed martensitic steel at an inter-critical annealing temperature. Moreover, Sugimoto et al. [25,26,27] reported that strengthening improved retained austenite characteristics, and improved impact properties were achieved via hot and warm forging of TBF steels.

The first main goal of the present paper is to map the phase transformations of a hot-rolled lean medium Mn steel in relation to different austenitizing conditions after hot-rolling and then to compare measured M_s_ temperatures as a function of austenitization temperature (T_γ_) as well as obtained prior austenite grain size (PAGS). This information can be successfully used by the hot rolling process and final annealing including the Q&P process of this steel group, the latter in both hot-rolled and cold-rolled condition. The second main goal of this paper is to construct a CCT (continuous cooling transformation) diagram for the given steel for its use in the subsequent heat treatment for the evaluation of the resulting microstructure at individual cooling rates. Finally, the constitution of a DCCT (deformation continuous cooling transformation) diagram allows for the prediction of transformational behavior in the present steel during hot-rolling. The paper predicts the behavior of the phase transformations during hot rolling and provides a comprehensive view of the retained austenite amount after the process of continuous cooling with and without deformation. It also gives results on the samples hardness that have been subjected to deformation compared to undeformed samples.

## 2. Materials and Methods

The base material used in the present experiments is a lean medium Mn Q&P steel foreseen to be used mostly in the automobile industry. This material was cast into ingots of 80 kg under laboratory conditions in a medium frequency furnace; the ingot dimensions were 80 × 120 × 1000 mm^3^, followed by hot rolling to a final thickness of 6 mm. The chemical composition of experimental samples was verified by a spectral analyzer Q4 TASMAN (Bruker BioSpin GmbH, Ettlingen, Germany) and it is presented in Table 1. The experimental steel is alloyed 1.4 wt.% of Si to suppress the formation of cementite in the microstructure allowing sufficient enrichment of the austenite by C. Metallographic samples were etched by a 3% Nital etchant (3% HNO_3_ + 97% ethanol). A different approach was used to prepare the samples for grain size measurement. The quenched specimens after deformation were sliced along the axial section. The sections were polished and etched in a solution of picric acid (5 g) + H_2_O (100 mL) + HCl (2 mL) + benzene sulfonic acid (2 g) at 50 °C for 3–4 min. Then, the optical micrographs were recorded in the center region of the samples using a metalloscope (Neophot32, Carl Zeiss GmBH, Jena, Gemany), and the average primary austenite grain size of specimens was measured using the software Image-Pro Plus (Media Cybernetics, Rockville, MD, USA) according to the line interception method described in ASTM E112-96 standards.

The hot-rolled samples were used for the dilatometric measurements related to the influence of the T_γ_ on the microstructural development and evolution of PAGS. Allowed test sample shape is a cylinder with dimensions ø 4 × 10 mm [28,29]. An important requirement for the sample preparation is to meet the perpendicularity of the face to the longitudinal axis of the sample.

Selected thermal modes are depicted in Figure 1. The heating of each sample was performed at the same rate of 1 °C/s. The steps followed are a holding period of 600 s at the specific temperature in a range of 800–1200 °C (see Figure 1) and cooling with a rate of 20 °C/s to establish completely martensitic structure. Afterwards, the Vickers hardness was measured while five measurements per condition were performed on each sample under HV1 loading [11].

Dilatometry was also used for the determination of phase transformations and construction of CCT and DCCT diagrams, whereby samples were tested using a Bähr DIL 805A dilatometer (TA Instruments, New Castle, DE, USA). The initial temperature was set at 950 °C with the holding time 10 min in order to ensure full austenitization. The heating rate was set to 1 °C/s. The samples were cooled at eight rates of 100, 10, 5, 1, 0.5, 0.1, 0.05, and 0.01 °C/s, respectively. In this regard, the given cooling rates were chosen to cover the full range of anticipated phase transformations in the investigated material [30]. Deformation of the sample occurred at the end of the holding stage at an austenitizing temperature of 950 °C and its rate was set to 0.1 s^−1^. The degree of deformation was set to 60%, i.e., the sample was compressed from 10 mm to 4 mm, followed by continuous cooling. Vickers hardness measurements were also performed on these samples equally to the previous case.

An accurate comparison of the amount of residual austenite in CCT and DCCT samples was determined by SMM, which is compared with data from XRD measurements. The XRD method is the most frequently used. Magnetization measurements have however intrinsic advantages as they probe the bulk of the materials. In magnetization measurements, the saturation magnetization can be obtained from the magnetization curve of a reference fully ferric or martensitic material. In this method, the amount of paramagnetic retained austenite can be measured due to a decrease of saturation magnetization compared to the fully ferromagnetic reference material [31,32].

After the magnetization measurements, samples with a diameter of 4 mm were cut along the transverse direction of the cylinder samples and polished for the XRD measurements. Very low force was applied during cutting and mechanical polishing in order to avoid stress-induced transformation of the retained austenite. The XRD measurements were performed on a Siemens’ X-ray diffractometer (Siemens, Aubrey, TX, USA) using CoKα radiation. A more detailed description of experimental conditions has been published in [33].

## 3. Results and Discussion

### 3.1. Determination of Ms Temperature

The transformation temperatures A_c1_ (austenite start temperature) and Ac_3_ (austenite finish temperature) were determined by a tangent method as shown in Figure 2. The values obtained from the dilatometric measurements for the lean medium Mn steel are Ac_1_ = 722 °C and Ac_3_ = 865 °C, respectively. Moreover, the graph is supplemented by a derivation curve of the given dilatation curve, which serves for the confirmation of the Ac_1_ and Ac_3_ temperature obtained by the tangent method.

Dilatation curves from the austenitization temperature of 800 °C, 1000 °C, and 1200 °C, are exemplarily presented in Figure 3a. Dilatation curves were used to determine the initial M_s_. Dashed lines in the figure are tangent line copying linear parts of the dilatation curves. The deflection point between the tangent line and the curve is considered as the Ms temperature. The final microstructures obtained for the individual heating modes, followed by the cooling with a rate of 20 °C/s, are shown in Figure 3b–d. All microstructures comprised of lath martensite with different PAGS and lath sizes dependent on the T_γ_. In this respect, it is evident that the PAGS and lath size increased with an increase in the T_γ_. The grain growth of PAGS also explains an increase of Ms temperature at elevated austenitization temperatures (Figure 3a).

The effect of PAGS was investigated more in detail based on dilatometric measurements obtained after the cooling of the individual samples with respect to the determined austenitization temperatures, similarly as in [34,35]. Figure 4a exemplarily highlights austenitic grain boundaries at 1100 °C that were processed using an ImageJ software to determine the average value from the PAGS distribution (Figure 4b). The obtained average PAGS values of the investigated steel and the corresponding M_s_ temperature are given in Table 2. We observe that as the austenitization temperature increases in the region of 800–1100 °C, the size of the individual grains increases almost linearly (Figure 5a). This was also observed in [36,37]. The grain size at 1100 °C was measured to be 33 μm. At the highest austenitization temperature of 1200 °C, there was however an abrupt increase of PAGS to 165 μm. Figure 5a,b represent the evolution the of PAGS and Ms temperature with austenitization temperature, respectively. It is evident that both evolutions can be well described by an exponential function.

### 3.2. Microstructure

Microstructures of the investigated lean medium Mn steel using SEM [38] for all cooling conditions are depicted in Figure 6. In a temperature range of 900–1200 °C, the microstructure consists of tempered and fresh martensite whereas at 800 °C the microstructure contains some ferrite and fresh martensite. The austenitization temperature is an important factor in the formation and size of the final martensitic grains. It can be observed that with an increase of austenitization temperature PAGS also increases. Thus, the largest primary austenite grains and the largest martensite lath size were formed by the sample with austenitization temperature of 1200 °C. Furthermore, it can be seen that in the martensitic laths for the austenitization conditions in a temperature at 900–1200 °C the needle-like carbides formed as a consequence of martensite self-tempering during cooling under the Ms-temperature. The amount of these carbides increases with an increase of applied austenitzation temperature as an aftermath of increased Ms temperature. In other words, the increase of Ms-temperature allows for more martensite self-tempering during cooling below this temperature.

### 3.3. Hardness

The Vickers hardness values HV1 are listed in Table 3 as a function of austenitization temperature. The results clearly demonstrate the decreasing hardness values in relation to an increase of austenitization temperature. The lower value of hardness 498 HV1 for the austenitizing temperature 800 °C compared to 900 °C is the result of this temperature being lower than the determined temperature Ac_3_ (865 °C). Therefore, only a partial austenitization occurs at 800 °C. Moreover, the sample austenitized at the lowest temperature of 800 °C has the largest uncertainty of measurement due to the fact that the microstructure consists of soft ferritic and hard martensite phase and thus, it manifests the largest heterogeneity of the microstructure. On the contrary, the samples annealed in the temperature range of 900–1200 °C exhibited a lower uncertainty of measurement compared to the sample austenized of 800 °C because of more homogeneous structure consisting of a mixture of fresh and tempered martensite. Furthermore, the hardness of the samples decreased from 511 HV at 900 °C to 447 HV1 at 1200 °C as a consequence of a larger amount of self-tempered martensite and the grain coarsening of the PAGS with temperature.

### 3.4. Comparison of CCT and DCCT Diagram

The three selected dilatation curves for 10 °C/s, 0.5 °C/s, and 0.05 °C/s for the CCT and DCCT cooling regimes are exemplarily depicted and compared in Figure 7, respectively. Figure 7a shows the dilatation curve at a cooling rate of 10 °C/s, whereby it is clear that the Ms temperature for this cooling rate is at 375 °C. Comparing Figure 7b, we can observe a significant difference in the resulting structure due to the deformation of the sample prior to the process of controlled cooling. The small derivation from the linearity at 409 °C for the deformed sample is related to the start of the bainitic transformation Bs, which was also confirmed by the derivation of the given dilatation curve. This was followed by the martensitic transformation, which began at 390 °C. By reducing the cooling rate to 0.5 °C/s, (Figure 7c), we observe the onset of the bainitic transformation at 500 °C. The Ms temperature decreased to 337 °C compared to the previous cooling curve in Figure 7a. Furthermore, the ferritic transformation Fs also occurred, but it is poorly observable in the given dilatation curve. Therefore, its precise determination is supported by the derivation of the dilatation curve from which it is possible to determine the beginning of this transformation. The temperature, at which the ferritic transformation started, is 620 °C. In Figure 7d, there is a dilatation curve after deformation at a cooling rate of 0.5 °C/s, whereby in total of three subsequent phase transformations can be observed. The ferritic conversion of Fs was the first to be recorded at 581 °C, while the end of this transformation Ff could be observed at 551 °C. Subsequently, the bainitic transformation continued at 538 °C. Finally, the martensitic transformation started at 353 °C. Comparing Figure 7c,d with each other, we observe the same sequence of the resulting phase transformations, with a slight difference in their initial temperatures. A decrease in the cooling from 0.5 °C/s (Figure 7c,d) to 0.05 °C/s (Figure 7e,f) resulted in a rise of the Fs temperature for both CCT and DCCT regime. Moreover, a clear tendency of a decrease of cooling rate on the onset of bainitic transformation could not be observed. For the CCT regime, the Bs temperature increases with a deceleration of cooling rate, whereas in the DCCT mode a decrease of Bs could be detected. The start of the martensitic transformation, represented by the Ms temperature, could not be observed in the given dilatometric curves for the lowest cooling rate. Therefore, all these dilatation curves must be compared with the resulting microstructures for the accurate evaluation of underlying phase transformation, which will be discussed later on.

Comparison of CCT and DCCT diagram (Figure 8) of investigated Mn steel was constructed based on the data from all eight aforementioned measured cooling curves in the range of cooling rates from 100 °C/s to 0.01 °C/s per condition. The comparison diagram clearly defines the areas of the ferritic, bainitic, or martensitic formation. Moreover, there also exists a small region of pearlite transformation in the present steel but related to very low cooling rates applicable for any practical use in real heat treatment.

The red lines in the diagram represent the Ac_1_ and Ac_3_ temperatures, which delimit the intercritical region during the heating stage.

It is visible in the CCT diagram that the Ms temperature decreases from 386 °C at 100 °C/s to about 100 °C for 0.05 °C/s. The Ms temperature is not determined for 0.01 °C/s because of the fact that the physical limit of the applied dilatometer, with respect to its minimal temperature during cooling, is 50 °C. The Ms temperature from the DCCT diagram is very similar to the one from the CCT diagram in a range of cooling rates from 100 °C/s to 0.5 ° C/s. In this range, the Ms temperature of DCCT diagram was about 10 °C lower compared to the one from the CCT diagram.

In contrast to the CCT diagram, in the DCCT diagram a heterogeneous structure was formed consisting mainly of martensite and a small amount of bainite already at the cooling rates of 100 °C/s and 10 °C/s. In the case of these two cooling rates, the bainitic phase represented only a very narrow region above the onset of martensite formation and began at 410 °C. As the cooling rate gradually decreased, the region of the bainitic transformation expanded. When comparing the bainitic region of both diagrams, it is clear that in the DCCT diagram this area is present in all measured cooling rates, while in the classical CCT diagram, the bainitic region does not begin to occur until a cooling rate of 5 °C/s.

At the cooling rate of 1 °C/s in DCCT diagram, a small amount of ferrite starts to form at a temperature of about 500 °C. The ferritic structure occurs up to the last cooling rate of 0.01 °C/s. In the CCT diagram, the ferritic transformation started first at a cooling rate of 0.5 °C/s. The temperature Fs in the CCT diagram is significantly higher by about 40 °C at a cooling rate of 0.5 °C/s up to 60 °C at a cooling rate of 0.1 °C compared to the DCCT diagram. In general, it can be stated that the ferritic and bainitic transformation in the DCCT diagram are shifted to faster cooling rates compared to the CCT diagram.

The final microstructural constituent that occurred in both diagrams was pearlite. Its formation was recorded at the slowest cooling rate of 0.01 °C/s. It is obvious that the area of pearlitic transformation has a similar shape, but it is slightly larger in DCCT compared to CCT diagram. Obviously the cooling curve of 0.01 °C/s is the slowest cooling rate in both CCT and DCCT diagrams. Since in this case the measurement lasted more than one day, it can be stated that further measurements with a lower cooling rate are senseless due to no use in practice.

Based on the presented diagrams, it can be stated that the deformation enhances the overall diffusion intensity [39]. After deformation, the ferrite and bainite start temperatures move toward higher cooling rates and the phase areas (ferrite and pearlite) are wider compared to the undeformed material [40].

It can be concluded that the resulting CCT diagram can serve as a starting point for further heat treatment design of the present lean medium Mn steel, e.g., for the design of its final Q&P treatment, whereas the DCCT diagram can predict the phase transformations during hot rolling.

### 3.5. Comparison of Microstructural Evolution in CCT and DCCT Diagram

The microstructural investigation was performed for every investigated cooling rate used in the CCT and DCCT diagram of the present steel. All acquired microstructures are depicted in Figure 9a–p.

First of all, each investigated microstructure contained a certain amount of retained austenite, which will be discussed more in detail later on.

The microstructure of the CCT samples evolved from the fully martensitic (expect a small amount of retained austenite) for 100 °C/s and 10 °C/s (Figure 9a,b) to the structures containing except martensite and retained austenite a mixture of lower and upper bainite. This holds true for the cooling rate 5 °C/s and 1 °C/s (Figure 9c,d) with an increasing amount of bainite with a decrease in cooling rate. From the cooling rate of 0.5 °C/s (Figure 9e,f), ferrite started to appear in the microstructure, again, with an increasing amount, when the cooling rate decreased. Finally, the pearlite started to form at the slowest cooling rate of 0.01 °C/s (Figure 9h). In the case of the DCCT samples, the bainite structure began to appear next to martensite and retained austenite already at the highest cooling rate of 100°C/s (Figure 9i). Moreover, the ferritic transformation was accelerated and appeared already at a cooling rate 1 °C/s (Figure 9l–p). The pearlitic transformation could again be observed for the slowest cooling rate (Figure 9p). Furthermore, a small amount of fresh martensite could also be formed at the slowest cooling rate despite the fact that it could not be detected by dilatometry. This implies that the Ms temperature at the slowest cooling rate had to be above the room temperature. In general, microstructure of the DCCT samples is finer due to the grain refinement induced by deformation.

### 3.6. Determination of Phase Fraction

Figure 10 illustrates the determination of phase fraction from the dilatometric curves measured at the cooling rate of 0.05 °C/s in both CCT and DCCT mode by the lever rule. Table 4 summarizes the phase fractions for all cooling rates in both above-mentioned modes. It is evident that the deformation suppresses the progression of the ferritic and bainitic formation in the range of cooling rates (1 °C/s–0.05 °C/s), while the pearlitic transformation is slightly enhanced. However, the amount of pearlite at the slowest cooling rate remains in general very low. At the high cooling rates (>5 °C/s) the formation of bainite is accelerated by deformation and the lowest cooling of 0.01 °C/s diminishes the differences among individual structural compounds due to the sufficient time available to approach the equilibrium conditions. This is obvious from our CCT and DCCT diagram (Figure 8), where the onset of transformations is shifted to shorter times. On the one hand, deformation can accelerate phase transformations by inducing a higher density of dislocations, which increases the nucleation rate for the given transformation. On the other hand, the excess of dislocations can decelerate the growth stage, since the dislocations may act as obstacles against the propagation of ferrite and bainitic laths. Our results are in accordance with the findings of A. Basuki et al. [41], where the deformation led to deceleration of phase transformation in hot rolled TRIP steels.

### 3.7. Comparison of Retained Austenite Content in CCT and DCCT Diagram

Based on the obtained microstructures after final cooling, the amount of retained austenite was measured by the SMM, which was compared with the XRD method. The results of this analysis are shown in Table 5. As a result, it can be seen that a reduction in the cooling rate leads to an increase in the volume fraction of retained austenite for both the CCT and DCCT samples. This is due to the fact that a larger amount of ferrite and bainite is formed by lower cooling rates, which results in more efficient C partitioning from these structural compounds into austenite. This in turn stabilizes the retained austenite by the room temperature. Moreover, at the slowest cooling rate, there was a slight decrease in the retained austenite content for both the CCT and DCCT regime. This is associated with the pearlitic transformation, since pearlite is a mixture of ferrite and cementite. Cementite contains a high C content (6.68 wt.%) and therefore less C can partition into remaining austenite, resulting in its lower stabilization and subsequent partial transformation to fresh martensite. The amounts of retained austenite in the CCT and DCCT regime are quite comparable except the two highest cooling rates, 100 °C/s and 10 °C/s, respectively. At these cooling rates, a higher amount of retained austenite was measured for the DCCT regime compared to the CCT one. This is a consequence of the formation of a small amount of bainite in the case of the DCCT regime at these high cooling rates. Since Si prohibits the formation of cementite in this type of bainite, known as carbide-free bainite, more C can partition into remaining austenite, resulting in this more efficient stabilization in the DCCT mode compared to the CCT one. This finding is mainly due to the measurement accuracy of the SMM method, in which the resulting value of the residual austenite is measured from the entire volume of the experimental material, compared to the XRD method, the result of which is measured only at a surface sample area. In the XRD method, part of the residual austenite can transform during the mechanical preparation of the metallographic sample. Furthermore, when removing the upper part of the sample by grinding and polishing, we also remove certain a part of the hydrostatic pressure, which stabilizes austenite and so a certain part of it can transform into martensite [42]. Therefore, the XRD method leads to a slightly lower amount of determined retained austenite, which can be also seen in Table 5.

### 3.8. Comparison of Hardness in CCT and DCCT Diagram

The next step of the work was the analysis of the steel hardness after the dilatometric tests. In this respect, hardness measurements are compared in Table 6 and Figure 11 for each applied cooling rate from CCT and DCCT samples. It is obvious that hardness decreases with a decrease of cooling rate which confirms numerous data from literature [43,44].

The hardness changes together with different cooling rates, i.e., higher cooling rates applied during the cooling increase the hardness of the material. The reason for this is the formation of a larger amount of harder microstructural compounds such as martensite and bainite, when the cooling rate increases. The hardness of the material subjected to the deformation is slightly higher compared to non-deformed samples. This can be assigned to an evident grain refinement after deformation. Finally, it can also be noted that the uncertainty of measurements tends to increase with a decrease of cooling rate, since the microstructural changes from single phase martensitic to multiphase one result in a more pronounced heterogeneity.

## 4. Conclusions

The paper describes austenite decomposition of a lean medium Mn steel is mainly suitable for the Q&P process using dilatometry. The investigation of transformation behavior of the steel was performed using the five different austenitization temperatures in the range of 800–1200 °C by a constant cooling rate of 20 °C/s. The next part of the present contribution focused on the construction of the CCT and DCCT diagram for the investigated steel by application of the eight cooling rates in the range 100 °C/s–0.01 °C/s.

The following conclusions can be drawn from the present work:An increase of the austenitization temperature has increased both the grain size and martensitic laths size of final, predominantly martensitic microstructure, respectively. The microstructure contained ferrite and martensite for 800 °C, while it was martensitic in a temperature range of 900–1200 °C. Thus, in this temperature range, the full austenitization of the steel could be achieved. Furthermore, martensite tends to show an increased self-tempering M_s_ temperature increase due to elevated austenitization and therefore more time is available for this phenomenon to take place during the cooling stage.After deformation, the ferrite and bainite start temperatures accelerate toward higher cooling rates and the phase transformation regions of ferrite and pearlite become wider when compared to the undeformed material. However, the amount of ferrite and bainite decreases especially at intermediate cooling rates due to a deceleration of growth phase induced by deformation.The hardness of the samples subjected to the deformation is slightly higher compared to non-deformed samples. The reason for this is an evident grain refinement after deformation, which further increases the strength of the steel. The obtained hardness corresponds to the microstructural evolution detected from the dilatation curves as well as from the resulting microstructures.The volume fraction of retained austenite increased with a decrease of cooling rate as a result of a larger volume fraction of ferrite and bainite. This led to more pronounced partitioning of C into austenite, resulting in its better stabilization at room temperature. In case that pearlite was formed, a lower amount of retained austenite could be stabilized, because the large C content remained to be captured in the cementite.The volume fraction of retained austenite was comparable for the deformed and undeformed state except the two highest cooling rates, where due to deformation a small amount of carbide-free bainite could form.The SMM measurements were compared with the XRD method. The latter led to a slightly lower amount of retained austenite as a consequence of martensitic transformation induced by deformation stemming from the mechanical sample preparation.The resulting CCT diagram can serve as a starting point for further heat treatment design of the present lean medium Mn steel, e.g., for the design of its final Q&P treatment. Moreover, the DCCT diagram can help to predict the phase transformation behavior of the present steel during hot rolling.

## Figures and Tables

**Figure 1 materials-15-01753-f001:**
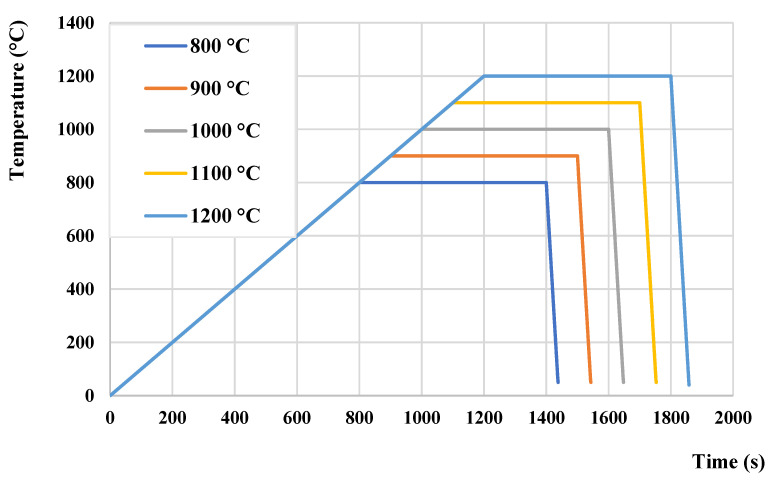
Thermal modes with a variation of austenitization temperature used for dilatometric analysis.

**Figure 2 materials-15-01753-f002:**
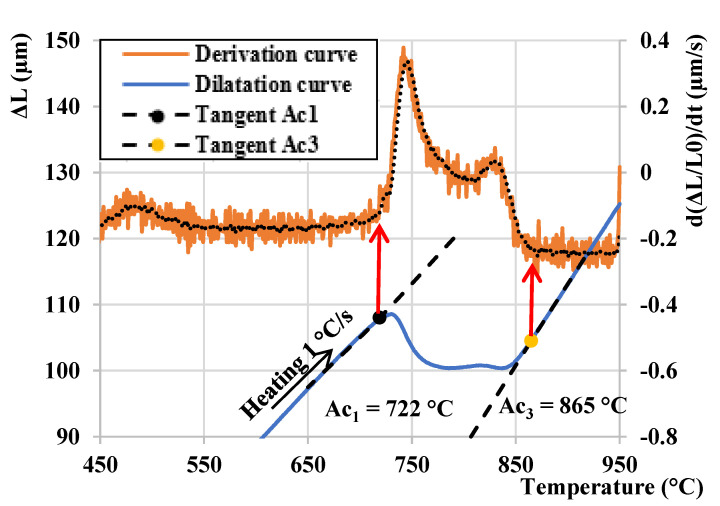
Dilatation curve for heating to 900 °C—determination of temperature Ac_1_ and Ac_3_.

**Figure 3 materials-15-01753-f003:**
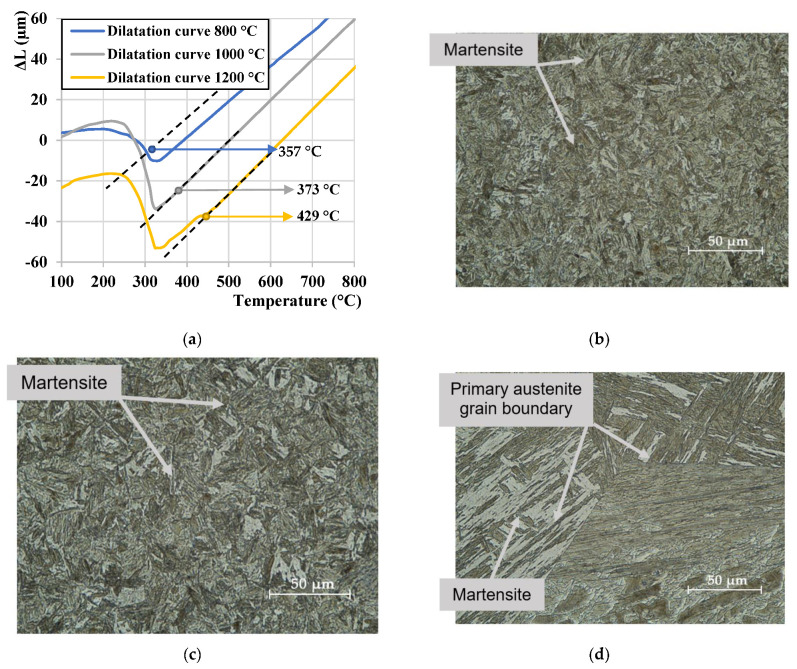
(**a**) Ms temperature for cooling mode from the austenitization temperature of 800 °C, 1000 °C, and 1200 °C obtained by dilatometry; LOM micrographs for the samples austenitized at (**b**) 800 °C; (**c**) 1000 °C; (**d**) 1200 °C.

**Figure 4 materials-15-01753-f004:**
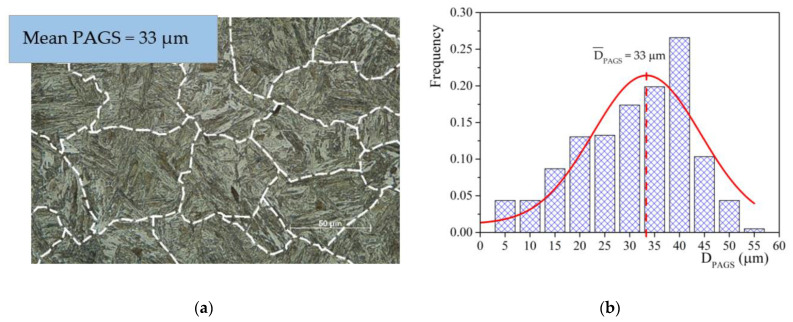
(**a**) Austenite grain boundaries determined by ImageJ at 1100 °C, (**b**) PAGS (prior austenite grain size) distribution calculation of its average value at austenitization temperature of 1100 °C.

**Figure 5 materials-15-01753-f005:**
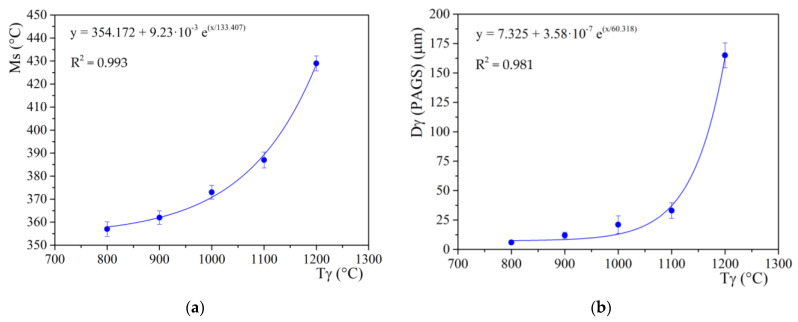
(**a**) Evolution of average PAGS and (**b**) Ms temperature with as a function of T_γ_.

**Figure 6 materials-15-01753-f006:**
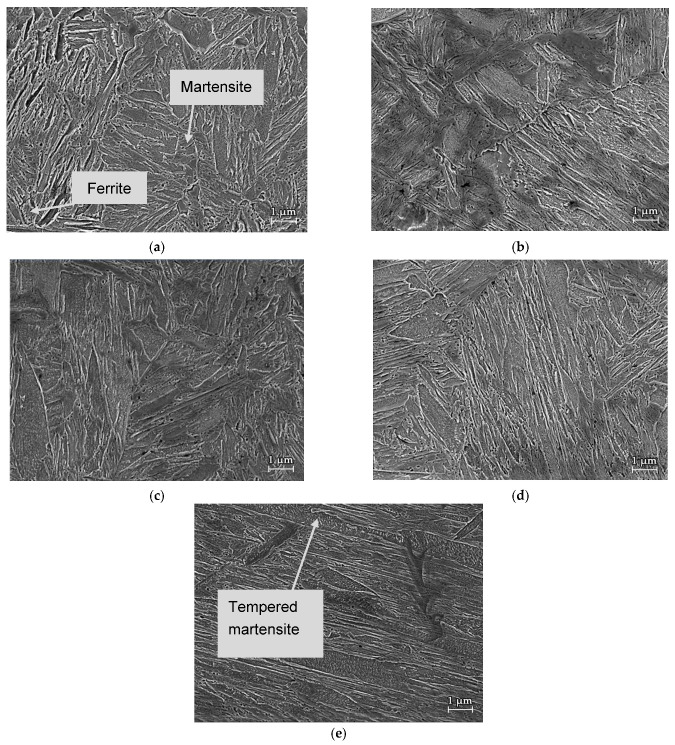
Microstructure of specimens obtained by SEM: T_γ_ (**a**) 800 °C, (**b**) 900 °C, (**c**) 1000 °C, (**d**) 1100 °C, (**e**) 1200 °C.

**Figure 7 materials-15-01753-f007:**
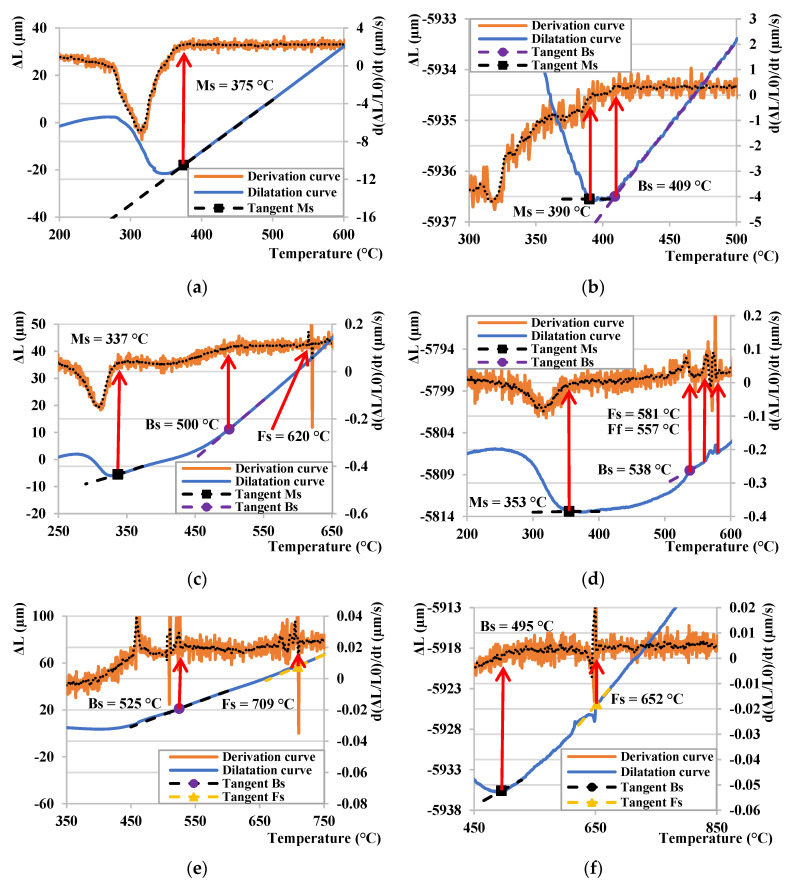
Comparison of dilatation curves for a cooling mode: (**a**) CCT 10 °C/s, (**b**) DCCT 10 °C/s, (**c**) CCT 0.5 °C/, (**d**) DCCT 0.5 °C/s, (**e**) CCT 0.05 °C/s, (**f**) DCCT 0.05 °C/s.

**Figure 8 materials-15-01753-f008:**
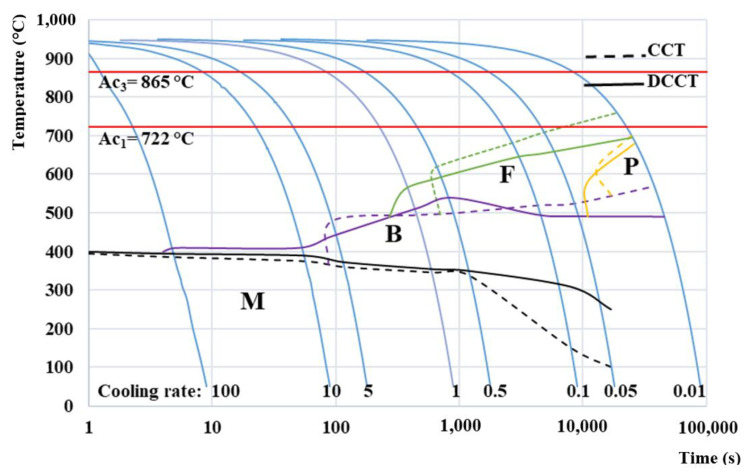
CCT and DCCT diagram of lean medium Mn 0.2C3Mn1.5Si steel.

**Figure 9 materials-15-01753-f009:**
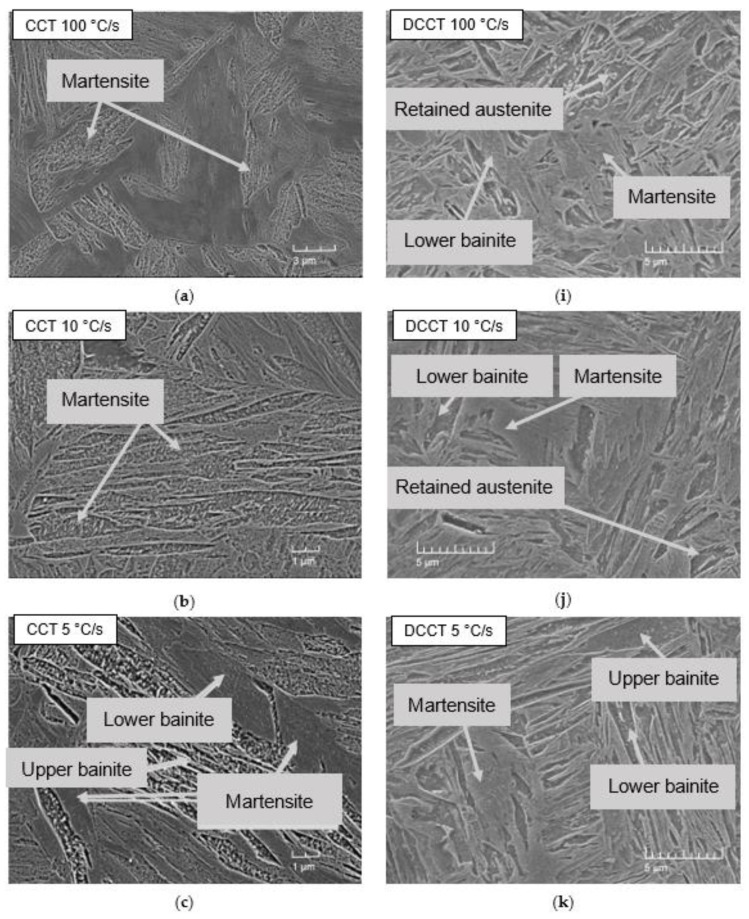
Comparison of SEM micrographs: (**a**) CCT 100 °C/s, (**b**) CCT 10 °C/, (**c**) CCT 5 °C/s, (**d**) CCT 1 °C/s, (**e**) CCT 0.5 °C/s, (**f**) CCT 0.1 °C/, (**g**) CCT 0.05 °C/s, (**h**) CCT 0.01 °C/s, (**i**) DCCT 100 °C/s, (**j**) DCCT 10 °C/, (**k**) DCCT 5 °C/s, (**l**) DCCT 1 °C/s, (**m**) DCCT 0.5 °C/s, (**n**) DCCT 0.1 °C/s, (**o**) DCCT 0.05 °C/s, (**p**) DCCT 0.01 °C/s.

**Figure 10 materials-15-01753-f010:**
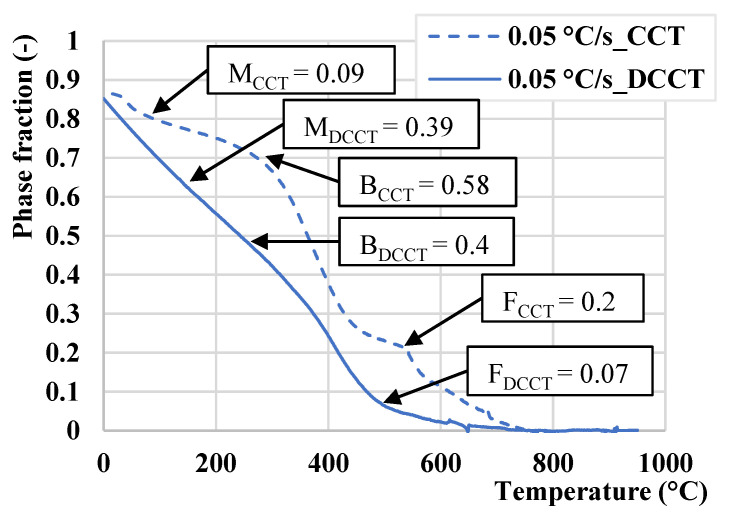
Determination of phase fraction from the dilatometric cooling curves in CCT and DCCT mode by lever lure for 0.05 °C/s.

**Figure 11 materials-15-01753-f011:**
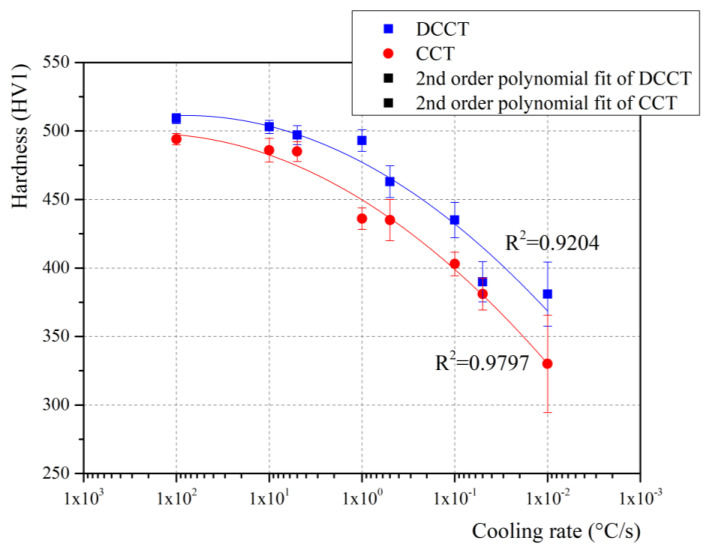
Hardness results for the CCT and DCCT samples cooled at different cooling rates.

**Table 1 materials-15-01753-t001:** Chemical composition of the investigated lean medium Mn Q&P steel (wt.%).

Element	C	Mn	Si	P	S	Al	Fe
Spectral analysis	0.19	2.96	1.46	0.001	0.001	0.04	balance
	±0.01	±0.04	±0.02	±0.00	±0.00	±0.00	±0.07

**Table 2 materials-15-01753-t002:** Ms (martensite start temperature) and PAGS at investigated austenitization temperatures.

T_γ_ (°C)	800	900	1000	1100	1200
Ms (°C)	357	362	373	387	429
D_PAGS_ (μm)	6 ± 1	12 ± 3	21 ± 5	33 ± 6	165 ± 18

**Table 3 materials-15-01753-t003:** Dependence of hardness HV1 (Vickers hardness, test force 10N) on austenitization temperature.

Tγ (°C)	800	900	1000	1100	1200
HV1	498 ± 18	511 ± 7	497 ± 4	473 ± 12	447 ± 8

**Table 4 materials-15-01753-t004:** Phase fractions determined from CCT and DCCT diagram.

dT/dt (°C/s)		100	10	5	1	0.5	0.1	0.05	0.01
CCT (vol.%)	Martensite	96	94	90	68	55	19	9	-
	Bainite	-	-	4	23	31	51	58	60
	Ferrite	-	-	-	-	4	18	20	25
	Pearlite	-	-	-	-	-	-	-	5
	Retained austenite	4	6	6	9	10	12	13	10
DCCT (vol.%)	Martensite	92	90	82	81	75	55	39	6
	Bainite	3	4	6	10	11	28	40	49
	Ferrite	-	-	-	2	3	5	7	23
	Pearlite	-	-	-	-	-	-	-	11
	Retained austenite	5	6	7	9	11	12	14	11

**Table 5 materials-15-01753-t005:** Comparison of retained austenite content.

dT/dt (°C/s)	100	10	5	1	0.5	0.1	0.05	0.01
CCT-RA-SMM (vol.%)	4.31 ±0.21	5.62 ±0.23	6.71 ±0.25	9.65 ±0.27	10.62 ±0.28	11.93 ±0.31	12.97 ±0.32	10.11 ±0.28
CCT-RA-XRD (vol.%)	2.91 ±0.17	4.25 ±0.19	4.88 ±0.20	7.29 ±0.23	8.65 ±0.24	10.02 ±0.26	11.87 ±0.27	8.25 ±0.24
DCCT-RA-SMM (vol.%)	7.90 ±0.23	6.12 ±0.27	6.31 ±0.27	9.07 ±0.28	10.05 ±0.29	12.22 ±0.31	14.11 ±0.34	11.5 ±0.31
DCCT-RA-XRD (vol.%)	6.23 ±0.22	5.89 ±0.23	6.02 ±0.24	7.89 ±0.25	8.29 ±0.27	9.68 ±0.28	12.23 ±0.30	8.32 ±0.27

**Table 6 materials-15-01753-t006:** Dependence of hardness HV1 on cooling rate in CCT and DCCT regime.

dT/dt (°C/s)	100	10	5	1	0.5	0.1	0.05	0.01
CCT HV1	494 ±4	486 ±9	485 ±7	436 ±8	435 ±15	403 ±9	381 ±12	320 ±35
DCCT HV1	509 ±4	503 ±5	497 ±7	493 ±8	463 ±12	435 ±13	390 ±15	381 ±23

## Data Availability

All the data is available within the manuscript.

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
