# Peer review of "Austenite Decomposition of a Lean Medium Mn Steel Suitable for Quenching and Partitioning Process: Comparison of CCT and DCCT Diagram and Their Microstructural Changes"

_materials, 2022, doi:10.3390/ma15051753_

Round 1

Reviewer 1 Report

The manuscript presents an interesting study about the phase transformations of a hot-rolled lean medium Mn steel in relation to different austenitizing conditions after hot-rolling. Also, in the study was construct a CCT and DCCT diagram. However, the paper needs minor revisions before it is processed further, some comments follow:

The abstract must be improved. The abstract is written qualitatively. The majority of the qualitative statements should be modified for quantified result comparisons. Please introduce the main conclusions or interpretations.

The introduction section must be improved. In the introduction section, a comprehensive and exhaustive review of the state of the art in the field of the study must be provided. Please refer to previous works, and highlight the experiments and results published previously. In the current form, the introduction section only provides basic/general information.

Even if the meaning of the acronyms is written in the abstract, please, also, introduce it in the manuscript text body.

Multiple citations have been introduced in bulk form "[1-3]", "[16-20]", "[25-29]" and not distributed in the text following the affirmations that must be supported. Please introduce citation at a specific position to assure a clear correspondence between the affirmations from the introduction section and the previous publication. Moreover, to avoid this type of citing, please cite review type of studies.

Please check carefully the correlation between the cited papers and the position of that reference in the manuscript text body.

Also, there are too many self-citation. Please keep only the necessary references.

Materials and Methods Section

Table 1 – Please check your experimental results. The Fe is missing from the table.

Figure 1 – the measuring units writing is inconsistent. Please introduce or remove the blank space between the value of the temperature and the measuring unit.

Figure 1 – The Gridlines of the figure doesn’t allow a clear read of the maintaining or cooling time. Please improve.

Results and discussions Section

Figure 2 – Please provide a smoother curve for the Derivation curve. Please follow the example from Figure 7.

Figure 3 and Figure 4 – Please introduce Figure labels to highlight the areas of interest for the reader.

Figure 9 – The phase identification seems approximate, as there are no results to support the evaluation of the microstructure. Please introduce additional experiments to confirm the affirmations related to figure 9. Please provide elemental mapping (EDX), EBSD, phase evaluation by XRD or another suitable method. See: Doi: 10.1016/j.msea.2021.141183 and Doi: 10.1016/j.corsci.2010.07.023.

Author Response

We would like to thank all reviewers for their valuable comments, which led to a further improvement of our manuscript. All remarks of the reviewers have been addressed and the amendments are highlighted in the manuscript in yellow.

We would like to thank the reviewer for his comment. The dilatometric cooling curves for both CCT and DCCT mode were analyzed by a lever rule in order to obtain a quantitative determination of individual phase fractions. This analysis can be found in (tab. 4).

We hope that our amendments are satisfactory for all reviewers, leading to a final acceptation of our manuscript.

Thank you very much.

Best regards, Maros Eckert.

Reviewer 2 Report

Comments to the article

Table 1 Give the accuracy of determining the chemical composition

Line 68 When using the TRIP abbreviation for the first time, enter its full name.

Line 78 When using the PAGS abbreviation for the first time, enter its full name.

Line 86 Does the tested steel Mn Q&P have a type designation 1.XXXX? If so, please provide them.

Figure 2 In the caption under the figure the text "Figure 2. Dilatation curve for heating 900 ° C for heating ..." - the text should be corrected

Figure 3 b, c and d - in the caption, add that it is a microstructure determined for a given temperature

Table 4 Give the accuracy of the determination of the residual austenite

Fig. 10 Add trend lines to the hardness test results 

Author Response

We would like to thank all reviewers for their valuable comments, which led to a further improvement of our manuscript. All remarks of the reviewers have been addressed and the amendments are highlighted in the manuscript in yellow.

We hope that our amendments are satisfactory for all reviewers, leading to a final acceptation of our manuscript.

Thank you very much.

Best regards, Maros Eckert.

Reviewer 3 Report

The present work is devoted to the study of heat treatments on medium Mn TRIP steel. The work is very interesting and the authors perform a lot of experimental work. I suggest publication for the work after the following revisions.

-Please improve the introduction section adding recent works regarding the effect of heat treatments of TRIP steels in order to have a comprehensive view of the present literature on the subject. In particular I can suggest: https://doi.org/10.3390/met11122055 and https://doi.org/10.3390/ma14020288

-Please add at the end of the introduction a sentence regarding the novelty of the work

-Please improve the experimental section adding proper explanation regarding the microstructural characterization: where the samples were taken, metallographic preparation, etch, instruments employed etc

-The amount of retained austenite played a key role in these steels. It should be interesting to compare the results obtained with magnetic methods with the ones obtained by XRD (the method that is suggested by international standards)

-Fig.9 is too big, I suggest to divide in more figures. Moreover, I suggest to remove the data bar in the end of the photo and let only the marker in order to make it more visible

-All hardness values must be presented with proper error bars and statistical errors (for example in Fig.10)

-At least for some selected samples also tensile test should be performed in order to confirm the mechanical behaviour suggested by the hardness tests

-The discussion of the results is quite poor and I suggest to improve it also on the base of the present literature comparing the obtained results with the ones reported in literature.

Author Response

We would like to thank all reviewers for their valuable comments, which led to a further improvement of our manuscript. All remarks of the reviewers have been addressed and the amendments are highlighted in the manuscript in yellow.

We would like to thank the reviewer for his valuable comment. We performed additional XRD experiments in order to determine the amount of retained austenite by this method. We compared them with the results obtained by SMM in Tab. 5. The XRD values were slightly lower compared to the SMM ones due to the following reasons: 1. XRD measured RA content locally while SMM is a bulk method, 2. some RA can transform during mechanical grinding and polishing into martensite 3. a decrease of hydrostatic pressure in the surface area decreases the mechanical stability of RA.

We would like to thank the reviewer for this comment. Since dilatometry is a high through-put method, it is rather common in literature to publish CCT and DCCT diagrams with hardness measurements only. In our case, we only have a small amount of material available for small dilatometric samples and there is no additional material left to perform further annealings and manufacturing of tensile samples. This will need a preparation of extra material and new heats, which is extremely cost- and time-consuming. Therefore, in this case, we cannot recently perform these additional tensile tests. However, we will keep this suggestion of the reviewer in mind for our further research.  

We hope that our amendments are satisfactory for all reviewers, leading to a final acceptation of our manuscript.

Thank you very much.

Best regards, Maros Eckert.

Round 2

Reviewer 3 Report

Considering that the authors have properly answered to the main issues of the first revision i suggest publication in its form for the present work